# Influence of SARS-COV-2 Infection on Cytokine Production by Mitogen-Stimulated Peripheral Blood Mononuclear Cells and Neutrophils in COVID-19 Intensive Care Unit Patients

**DOI:** 10.3390/microorganisms10112194

**Published:** 2022-11-04

**Authors:** Sahar Essa, Mohammed Shamsah, Abdalaziz H. Alsarraf, Ali Esmaeil, Ahmed Al-Shammasi, Raj Raghupathy

**Affiliations:** 1Department of Microbiology, Faculty of Medicine, Kuwait University, Kuwait City 46300, Kuwait; 2Department of Anesthesia, Critical Care and Pain Management, Al Adan Hospital, Ministry of Health, Kuwait City 47000, Kuwait; 3Ministry of Health, Adan Hospital, Kuwait City 47000, Kuwait

**Keywords:** SARS-COV-2, intensive care unit, peripheral blood mononuclear cells, neutrophils, cytokines, COVID-19 patients

## Abstract

We sought to investigate the influence of SARS-CoV-2 infection on the cytokine profiles of peripheral blood mononuclear cells (PBMCs) and neutrophils from coronavirus disease 2019 (COVID-19) intensive care unit (ICU) patients. Neutrophils and PBMCs were separated and stimulated with the mitogen phytohemagglutinin. Culture supernatants of mitogen-stimulated PBMCs and neutrophils from 88 COVID-19 ICU patients and 88 healthy controls were evaluated for levels of granulocyte-macrophage colony-stimulating factor (GM-CSF), interferon (IFN)-α, IFN-γ, interleukin (IL)-2, -4, -5, -6, -9, -10, -12, -17A, and tumor necrosis factor (TNF)-α using anti-cytokine antibody MACSPlex capture beads. Cytokine profiles of PBMCs showed significantly lower levels of GM-CSF, IFN-γ, IL-6, IL-9, IL-10, IL-17A, and TNF-α (*p* < 0.0001) in COVID-19 ICU patients. In contrast, COVID-19 ICU patients showed higher median levels of IL-2 (*p* < 0.001) and IL-5 (*p* < 0.01) by PBMCs. As for neutrophils, COVID-19 ICU patients showed significantly lower levels of GM-CSF, IFN-γ, IL-2, IL-4, IL-5, IL-6, IL-9, IL-10, IL-17A, IL-12, TNF-α (*p* < 0.0001), and IFN-α (*p* < 0.01). T-helper (Th)1:Th2 cytokine ratios revealed lower inflammatory cytokine for PBMCs and neutrophils in COVID-19 ICU patients. Cytokine production profiles and Th1:Th2 cytokine ratios suggest that severe acute respiratory syndrome coronavirus 2 (SARS-CoV-2) infection has an immunomodulatory effect on PBMCs and neutrophils. This study also suggests that the increased levels of several cytokines in the serum are not sourced from PBMCs and neutrophils.

## 1. Introduction

The coronavirus disease 2019 (COVID-19) pandemic has become the ultimate health challenge worldwide [1]. The array of clinical presentations of COVID-19 varies, causing asymptomatic, mild illness, acute respiratory distress syndrome (ARDS), cardiac pathology, or death [2,3]. Research has added to our knowledge of the viral immunopathogenesis of COVID-19 [2,3]. A vigorous immune response, including the production of several cytokines, is critical in fighting viral infections; however, cytokine secretion patterns may support or combat COVID-19 [4]. A case of death after the administration of the anti-COVID-19 vaccine suggests the involvement of vaccine administration in the cytokine storm [5]. The cytokines relevant to anti-viral immune defenses are those involved in initiating immune responses (e.g., interleukin (IL)-2 and IL-4), pro-inflammatory cytokines (e.g., interferon (IFN)-α, -γ, and -λ; IL-6 IL-17 and tumor necrosis factor (TNF-α)), and anti-inflammatory cytokines (e.g., IL-10) [6,7]. Various studies have reported abnormal serum levels of the following cytokines in COVID-19 intensive care unit (ICU) patients: IL-1, IL-2, IL-4, IL-6, IL-7, IL-10, IL-12, IL-13, IL-17, macrophage colony-stimulating factor (M-CSF), G-CSF, granulocyte-macrophage colony-stimulating factor (GM-CSF), IFN-γ, hepatocyte growth factor (HGF), TNF-α, and vascular endothelial growth factor (VEGF) [8,9,10,11]. A crucial event in severe acute respiratory syndrome coronavirus 2 (SARS-CoV-2) infection might be the reduction in antiviral defense mechanisms associated with innate immunity and the raised secretion of inflammatory cytokines [12]. Immunomodulation of cytokine responses by the virus has been documented to cause much of the COVID-19 pathophysiology, although the exact mechanisms are not yet entirely known [13].

Imbalance in the levels of the T-helper (Th) subsets (Th1/Th2/Th17) and regulatory T-cells (Tregs) is proposed to be involved in the disease severity and prognosis of COVID-19 [13]. Cluster of differentiation (CD)4^+^ T-cells are divided into diverse subtypes based on their cytokine secretion patterns, including Th1 cells (producing IFN-γ, IL-2, and TNF-α), Th2 cells (IL-4, IL-5, IL-9, IL-10, and IL-13), Th17 cells (IL-17, IL-22), and Treg (TGF-β, IL-10) [6].

Neutrophils are primary and vital players in natural immunity as they are among the earliest leukocytes to be mobilized during infections [14,15]. High neutrophil counts have been reported in the bronchoalveolar lavage fluid of COVID-19 patients with severe disease compared to those with moderate infection [9,16]. The histological changes in lung injury during COVID-19 pneumonia could be due to neutrophil extracellular traps (NET) and the immense accumulation of neutrophils in COVID-19 patients [10,17]. Viral nucleic acid and cytokines could stimulate the formation of NET [18]. Although the specific effect of NET on antiviral immunity is not yet understood, it is suggested to play a key role in the pathogenesis of COVID-19 [19,20]. A recent article suggested that pro-inflammatory cytokines secreted by neutrophils could contribute to the cytokine storm and the immunopathological complications observed in severely ill COVID-19 patients [21].

In this context, this study aimed to measure the cytokine profiles of PBMCs, neutrophils, and serum from COVID-19 ICU patients. This study sought to determine the effects of SARS-CoV-2 infection on cytokine production and to ascertain whether PBMCs or neutrophils contribute to serum levels of cytokines in COVID-19 patients.

## 2. Materials and Methods

### 2.1. Study Population

A statistical power analysis indicated a sample size of 75 subjects, assuming a 95% rate of change in COVID-19 infection, a 95% confidence interval, and a maximum accepted error of 0.05. The 88 COVID-19 patients included in this study were admitted to the intensive care unit (ICU) of Adan Hospital, Kuwait, according to the criteria of the Society of Critical Care Medicine (SCCM). Inclusion criteria were based on the detection of SARS-CoV-2 quantitative reverse transcriptase-polymerase chain reaction (RT-qPCR). Eighty-eight sex and age-matched healthy subjects were inducted from the local population through social media advertising. Nasopharyngeal swab samples were collected for respiratory investigations and blood samples were collected for laboratory investigations and isolation of PBMCs and neutrophils from each subject (COVID-19 patients and healthy controls). Healthy subjects were screened for SARS-CoV-2 by RT-PCR. Samples were collected between January and June 2021; the strains circulating globally during this period were mainly Alpha and Delta. Respiratory viral infections were detected as described previously [22]. Healthy controls with SARS-CoV-2 and respiratory viral infections were excluded from the study. Healthy control subjects were tested for Erythrocyte Sedimentation Rate (ESR) and C-Reactive Protein (CRP). Only subjects who were negative for these tests were included to eliminate those with active inflammatory conditions.

Eight mL of fresh blood was obtained from each subject (COVID-19 patients and healthy controls) and processed for mitogen-induced stimulation of PBMCs and neutrophil within one hour of collection. Samples were collected at the time of admission from COVID-19 ICU patients. Blood samples for laboratory investigations were subjected to the following tests: complete blood count (CELL-DYN 1700), liver function tests including ALT, AST, total bilirubin, direct bilirubin, CRP, lactate dehydrogenase (LDH), ESR, the pressure of arterial oxygen to fractional inspired oxygen concentration (PaO2/FiO2) ratio, Lactate, Troponin, and serum albumin (HITACHI 912 automatic analyzer, Roche Diagnostics, Mannheim, Germany), and α-fetoprotein (VIDAS autoanalyzer, BioMérieux, France). Clinical data also included underlining diseases, including hypertension, diabetes, asthma, and cardiovascular diseases. This research was started after obtaining ethical approval from the Human Research Committee of the Ministry of Health, Kuwait (No. 1410/2020).

### 2.2. Real-Time Reverse Transcriptase PCR Assays for Detection of SARS-CoV-2 RNA

Nasopharyngeal swabs were obtained from patients suspected of being infected with SARS-CoV-2. Total RNA was extracted by means of the Roche MagNA Pure LC system (Roche Diagnostics, Indianapolis, IN, USA) and the extracted RNA was tested for SARS-CoV-2 RNA by an in-house RT-PCR assay. Three genes, including E and N genes of SARS-CoV-2, and the human RNAse gene, were detected using the Quant Studio Real-Time PCR system (ThermoFisher Scientific, Waltham, MA, USA) and the SOLIScript 1-step multiplex probe kit (ROX) (SOLIS BIODYNE, USA), according to the manufacturer’s protocol. Identification of the E-genewas done for screening, and detection of the N-gene confirmed SARS-CoV-2 infection, while human RNAse was used as control [23]. A cycle threshold value less than 37 was documented as a positive result, and 38 and more was documented as a negative result.

### 2.3. Isolation of PBMCs

Peripheral blood mononuclear cells (PBMCs) were separated from fresh whole blood by density-gradient centrifugation on Ficoll-Hypaque (Pharmacia Biotech, Uppsala, Sweden) [24]. PBMCs were suspended at a final cell concentration of 8 × 10^6^ cells/mL in Roswell Park Memorial Institute (RPMI) medium (Gibco BRL, New York, NY, USA) which contained 10% fetal bovine serum (FBS) (Eurobio Scientific, Scandinavie Les Ulis, France).

### 2.4. Isolation of Neutrophils

A mixture of Dextran 500 and sodium metrizoate was used to isolate neutrophils from whole blood using commercially available separation media as described previously [24].

### 2.5. Mitogen-Induced Stimulation

Stimulation of PBMCs and neutrophils with the mitogen phytohemagglutinin (PHA) was reported by us previously [24]. Standardization trials demonstrated that optimal stimulation of PBMCs and neutrophils occurred at 96 h (data not shown).

### 2.6. Estimation of Cytokine Levels

Serum levels of cytokines as well as the levels of cytokines secreted by mitogen-stimulated PBMCs and neutrophils were estimated by a flow cytometry bead-based array with a MACSPlex Cytokine kit (Miltenyi Biotec, Bergisch Gladbach, Germany). The following cytokines were estimated: GM-CSF, IFN-α, IFN-γ, IL-2, IL-4, IL-5, IL-6, IL-9, IL-10, IL-12, and TNF-α. Samples were processed in a MACSQuant Analyzer 10 and evaluated using the Express Mode option of the MACSQuantify 2.8 software (Miltenyi Biotec, Bergisch Gladbach, Germany).

### 2.7. Statistical Analysis

Categorical variables are presented as frequencies and percentages compared using Chi-square or Fisher’s exact test. Continuous variables were reported as means and standard deviations (SDs) and compared to Student’s t-test. Univariate and multivariate analyses were performed to calculate the odds ratios (ORs) with their corresponding 95% confidence intervals (CI). Statistical significance was calculated using the Graph Pad Prism 9.1.1 software (GraphPad Software Inc.). The unpaired t-test was used to calculate p-values. Statistical analyses were two-sided; *p* < 0.05 was considered statistically significant. 

## 3. Results

### 3.1. Characteristics of the Patients Studied

A total of 88 (27 female, 61 male) COVID-19 ICU patients and 88 age- and sex-matched healthy controls were enrolled in this study. The mean age of the subjects was 59.9 years (range 34-85 years). Table 1 shows the gender and age of deceased and transferred COVID-19 ICU patients. As can be seen from Table 1, 10 (11.4%) passed away, and 78 (88.6%) were transferred from the ICU (OR = 0.97; 95% CI: 0.94–1.00). The difference in the number of deceased and transferred patients is statistically significant (*p* = 0.029). Of the deceased patients, 4 (40%) were female, and 6 (60%) were male. Of the transferred patients 23 (29.5%) were female and 55 (70.5%) were male with OR = 1.59; 95% CI: 0.41–6.18. The mean age of the deceased group was 60.9 years, and that of the transferred group was 52.8 years with OR = 0.95; 95% CI: 0.91–1.00. The difference in age between deceased and transferred patients was statistically significant (*p* = 0.112).

Multivariate analyses comparing the clinical features of deceased and transferred COVID-19 ICU patients are presented in (Table 2). Fever (*p* = 0.021), cough (*p* < 0.001), and other infections (*p* = 0.001) were significantly different between deceased and transferred COVID-19 ICU patients. On the other hand, pneumonia (*p* = 1.00) was not significantly different between deceased and transferred COVID-19 ICU patients.

No statistically significant differences were observed in laboratory investigations between deceased and transferred COVID-19 ICU patients (Table 3).

### 3.2. Comparison of Cytokine Levels Produced by PBMCs and Neutrophils from COVID-19 ICU Patients and Healthy Subjects (HC)

Significantly lower mean values were observed in the cytokine production profiles of PHA-stimulated PBMCs for GM-CSF, IFN-γ, IL-6, IL-9, IL-10, IL-17A, and TNF-α when compared to HC (Figure 1A,C,G–I,K,L, *p* < 0.0001). On the other hand, higher mean levels of IL-2 (Figure 1D, *p* < 0.001) and IL-5 (Figure 1F, *p* < 0.01) were produced by PHA-stimulated PBMCs from COVID-19 ICU patients compared to HC. No statistically significant differences were found in the levels of IFN-α, IL-4, and IL-12 of PHA-stimulated PBMCs compared with the HC group (Figure 1B,E,J, *p* > 0.05).

Cytokine levels of PHA-stimulated neutrophils from COVID-19 ICU patients and HC are shown in Figure 2. Statistically significant lower levels were detected for GM-CSF, IFN-α, IFN-γ, IL-2, IL-4, IL-5, IL-6, IL-9, IL-10, IL-17A, and TNF-α (Figure 2A,C–L; *p* < 0.0001 Figure 2B, *p* < 0.01).

### 3.3. Comparison of Cytokines Levels in Serum Samples of COVID-19 ICU Patients and Healthy Subjects (HC)

Cytokine production profiles of serum samples from COVID-19 ICU patients and HC are shown in Figure 3. Statistically significantly lower mean values of serum levels of IFN-γ, IL-5, IL-12, and IL-17A were seen in COVID-19 ICU patients as compared to HC (Figure 3C,J,K; *p* < 0.0001 Figure 3F, *p* < 0.01). In contrast, mean levels of GM-CSF, IFN-α, IL-2, IL-4, and IL-6 were higher in COVID-19 patients (Figure 3A; *p* < 0.001 Figure 3B,E,G; *p* < 0.0001 Figure 3D; *p* < 0.01). No statistically significant changes were seen in the serum levels of IL-9 and IL-10 (Figure 3H,I, *p* > 0.05).

### 3.4. Th1:Th2 Cytokine Ratios

The comparative levels of Th1:Th2 cytokine ratios are vital because these ratios can indicate the superiority of one cytokine pattern over the other. In 17 out of 24 combinations, a lower Th1 cytokine bias was observed in PBMCs from COVID-19 ICU patients (Table 4). For example, the ratios of TNF-α/IL-5, IFN-α/IL-10, IL-6/IL-5, TNF-α/IL-4, and IFN- γ/IL-5, were lower (≈24-, 23-, 20-, 10-, and 10-fold) in COVID-19 ICU patients as compared with HC subjects. A higher Th1 cytokine bias was observed in PBMCs of COVID-19 ICU patients for 7 out of 24 possible combinations. The ratios of IL-12/IL-4, IL-12/IL-10, IL-12/IL-5, IL-12/IL-9, IL-2/IL-10, and IL-2/IL-9 were substantially higher in COVID-19 ICU patients compared with HC subjects (≈5660-, 5333-, 2100-, 600-, 69-, and 24-fold differences, respectively).

A lower Th1 bias in cytokines produced by neutrophils was seen in 17 out of 24 possible combinations from COVID-19 ICU patients compared to HC (Table 4); the ratios of TNF-α/IL-5, IFN-α/IL-9, IFN-α/IL-4, IFN-α/IL-5, TNF-α/IL-4, and IL-6/IL-5 show lower differences in COVID-19 ICU patients versus HC individuals (≈21-, 20-, 16-, 5-, 4- and 4-fold respectively). In contrast, a more substantial Th1 cytokine bias is observed in 7 of 24 combinations of COVID-19 ICU patients compared with HC.

A greater Th1 cytokine bias is seen in 13 out of 24 combinations in the serum of COVID-19 ICU patients versus HC. For instance, the ratios of IL-2/IL-5, IL-2/IL-10, IL-2/IL-9, IL-6/IL-5, IL-6/IL-10 and IL-6/IL-9 were considerably higher in COVID-19 ICU patients as compared to HC subjects (≈8571-, 4800-, 4444-, 378-, 521-, and 172-fold differences respectively). Conversely, in 13 out of 24 combinations, a lower Th1 cytokine bias in the serum of COVID-19 ICU patients was detected. The ratios of IFN-γ/IL-4, IL-12/IL-4, IFN- γ/IL-9, TNF-α/IL-4, IFN- γ/IL-5 and IL-12/IL-9 were lower ((≈30-, 11-, 8, 4-, 4-, and 3- fold differences respectively) in COVID-19 ICU patients as compared to HC subjects.

## 4. Discussion

We hypothesized that SARS-CoV-2 infection modifies the responses of PBMCs and neutrophils and that this would be mirrored in the levels of cytokines produced by them. This hypothesis was reinforced by previous observations that SARS-CoV-2 infection triggers an increase in serum levels of cytokines [21]. Several studies have shown boosted inflammatory responses in COVID-19 patients [10,17,19]. This study was also aimed at ascertaining whether increased serum levels of cytokines are due, at least partly, to cytokine production by PBMCs and neutrophils.

Our results showed a statistically significant increase in the production of two cytokines, IL-2 and IL-5 by PBMCs and a statistically significant decrease in the production of seven cytokines GM-CSF, IFN-γ, IL-6, IL-9, IL-10, IL-17A, and TNF-α. Also, we detected a significant decrease in the secretion of twelve cytokines, GM-CSF, IFN-α, IFN-γ, IL-2, IL-4, IL-5, IL-6, IL-9, IL-10, IL-12, IL-17A, and TNF-α by neutrophils. On the other hand, serum levels of cytokines in COVID-19 ICU patients showed a significant increase in the production of GM-CSF, IFN-α, IL-2, IL-4, and IL-6 and a significant decrease in the levels of IFN-γ, IL-5, IL-12, IL-17A, and TNF-α.

Our results agree with early studies that reported elevated serum levels of GM-CSF in intensive and non-intensive COVID-19 patients [8,25]. These studies indicate that SARS-CoV-2 infection may contribute to the increased serum GM-CSF levels in patients with severe COVID-19. Moreover, these studies highlight that the source of GM-CSF and IL-7A is from memory cells, and our results suggest that the source of increased serum GM-CSF levels in COVID-19 patients is not PBMCs or neutrophils.

IFN-α is a type I interferon that can result in immunopathology during antiviral responses [26]. Hadjadj et al. documented reduced plasma levels of IFN-α in severe COVID-19 patients related to lower viral clearance [26]. IFN-γ, a type II interferon secreted by CD4+ T-cells, CD8+ T-cells, and natural killer (NK) cells [27,28], contributes to various immunological mechanisms, including antigen presentation, signal transduction, antibacterial, antiviral activities, and macrophage stimulation [29]. Increased secretion of IFNs is indicative of Th1-like cytokine response in COVID-19 patients. One strategy employed by the immune system to eliminate viral infections is to induce a Th1-dominated cytokine response [30]. Increased IFN-α and IFN-γ responses can lead to better outcomes in COVID-19 patients. By inhibiting the secretion of Th2 cytokines, IFN-γ alters the Th1/Th2 balance in favor of a Th2-like response. Elevated serum levels of IFN-γ have been previously described in patients with Middle East Respiratory Syndrome (MERS) [31]. Data from an animal model of SARS-CoV-2 infection confirmed that SARS-CoV-2 inhibits the induction of IFN-α and IFN-γ [12]. A recent study showed that serum levels of IFN-γ were significantly increased in COVID-19 patients compared with healthy subjects [32]. Another study showed that levels of IFN-γ, IL-6, and IL-10 were increased in COVID-19 patients but did not differ between ICU and non-ICU patients [33]. In addition, this study reported lower levels of IFN-γ in CD4+ T cells of COVID-19 patients with severe symptoms compared to those with mild symptoms, suggesting that the infection may initially downregulate CD4+ and CD8+ T-cells, thereby decreasing IFN-γ production. Another study found lower IFN-γ secretion by CD4+ T cells in severe COVID-19 patients than in moderate patients [34]. However, other reports establish an overactivity of CD8+ T-cells [35]. In the present study, we detected a significant decrease in the levels of IFN-γ produced by PBMCs and neutrophils and in the levels of IFN-α by neutrophils; this indicates an immunomodulatory effect on the production of IFN-γ and IFN-α by PBMCs and neutrophils. On the other hand, we detected a significant increase in serum levels of IFN-α in COVID-19 ICU patients compared with HC. This suggests that the source of the increased serum IFN-α levels observed in COVID-19 ICU patients is not PBMCs or neutrophils.

IL-2 plays a critical role in antiviral immunity, primarily through its direct effects on adaptive immune responses, thereby helping the body fight infections [36,37]. IL-2 also stimulates the differentiation of naive CD4+ T cells into Th1, Th2, and Th17 lymphocytes, and increases the cytotoxic potential of NK cells and cytotoxic T cells against virus-infected cells [38]. Increased serum IL-2 levels have been detected in many patients with severe COVID-19 compared to subjects with mild disease [17,38,39]. In contrast, Spolski et al. did not find an increase in IL-2 expression with disease severity [38]. In our study, we detected a significant increase in IL-2 production by PBMCs but not by neutrophils, which may indicate that PBMCs play a role in the elevated serum IL-2 levels in ICU patients with COVID-19.

IL-4, IL-5, IL -9, and IL-10 are produced by Th2 and mast cells. Bot et al. reported increased IL-4 expression during influenza virus infection, which negatively affected CD8+ T cells [40]. Studies on COVID-19 patients with severe respiratory symptoms have shown elevated serum levels of IL-4 [8,11,41]. Our results show a significant increase in serum IL-4 levels in COVID-19 ICU patients; this suggests that the source of IL-4 is not neutrophils and proposes an immunomodulatory effect on IL-4 production by neutrophils.

An increase in serum levels of the Th2 cytokines IL-5 and IL-9 has been reported in COVID-19 ICU patients compared to HC [13]. In our study, we detected a significant increase in IL-5 production levels by PBMCs but not by neutrophils, which may indicate that PBMCs may be responsible, in part, for the elevated serum levels of IL-5 in COVID-19 ICU patients. As for IL-9 and IL-10, we identified a significant decrease in PBMCs and neutrophils but in the serum, the differences were not significant. IL-10 prevents the stimulation of pro-inflammatory cytokines and inhibits the expression of the molecules of the major histocompatibility complex, which play a vital role in cellular immunity [42]. Several studies have correlated elevated serum IL-10 levels with the severity and progression of COVID-19, and it has been described to have a likely predictive value on disease prognosis [8,17,34,41,43].

Numerous cell types secrete IL-6 in response to infection. It is an essential trigger of hematopoiesis, immunological and biological activities, and has potent pro-inflammatory effects during viral infections [44,45]. Elevated serum IL-6 levels have been documented in patients with COVID-19 and were related to the severity and poor prognosis of the disease [17,46,47]. In addition, serum levels of IL-6 were higher in patients who died of COVID-19 than in those who recovered [25]. Previous studies indicated that SARS-CoV-2 infection contributes to the increased serum levels of IL-6 in patients with severe COVID-19 [17,25,46,47]. Our study showed a significant decrease in IL-6 production by PBMCs and neutrophils, as well as a significant increase in serum levels in intensive care patients with COVID-19, compared with healthy controls. Our findings are in alignment with previous studies on serum IL-6 [17,46,47], but, moreover, they suggest that the source of elevated serum IL-6 levels in COVID-19 ICU patients is not PBMCs and neutrophils.

IL-17A, a pro-inflammatory cytokine that plays a vital role in immunity to microbial infections, is produced by Th17 cells, T-cytotoxic cells, NK cells, and lymphoid cells [48,49]. Normal IL-17A production has been reported in patients with COVID-19, with no statistically significant correlation between patients with severe and mild symptoms [50]. In contrast, Huang et al. described elevated serum IL-17A levels in COVD-19 patients and a cytokine storm correlated with poor prognosis [8]. In the current study, we detected a statistically significant decrease in IL-17A levels in serum, PBMCs, and neutrophils of COVID-19 ICU patients compared with HCs, indicating an immunomodulatory effect of SARS-CoV-2 on IL-17 production by PBMCs and neutrophils.

Our results show a statistically significant decrease in TNF-α levels in serum, PBMCs, and neutrophils of COVID-19 ICU patients compared with healthy controls, suggesting an immunomodulatory effect of SARS-CoV-2 on TNF-α production of PBMCs and neutrophils. TNF-α is a potent pro-inflammatory cytokine produced by T-cells, macrophages, and monocytes. It has been associated with infectious diseases and tumors [51]. An increase in serum levels of TNF-α has been correlated with the severity of COVID-19 [8,17,34]. One study described similar results in patients with COVID-19 and found an inverse association between serum TNF-α levels and T-cell counts [43]. In contrast, Wan et al. reported normal TNF-α levels in COVID-19 patients [50]. Our results show a statistically significant decrease in TNF-α levels in serum, PBMCs, and neutrophils of COVID-19 ICU patients compared with healthy controls, suggesting an immunomodulatory effect of SARS-CoV-2 on TNF-α production by PBMCs and neutrophils.

Th1:Th2 cytokine ratios reveal some interesting features. A lower bias of inflammatory cytokines is detected in 17 of the possible combinations for PBMCs and neutrophils from COVID-19 ICU patients. This is indicative of the dysregulation of cytokine networks brought about by SARS-CoV-2 infection.

## 5. Conclusions

Our results clearly show a statistically significant decrease in the levels of seven cytokines produced by lymphocytes, GM-CSF, IFN-γ, IL-6, IL-9, IL-10, IL-17A, and TNF-α and twelve cytokines, GM-CSF, IFN-α, IFN-γ, IL-2, IL-4, IL-5, IL-6, IL-9, IL-10, IL-17A, IL-12, and TNF-α produced by neutrophils. These data suggest that the source of the cytokine storm in terms of GM-CSF, IFN-γ, IL-6, IL-9, IL-10, IL-17A, and TNF-α in ICU patients is not PBMCs. Similarly, the source of high serum cytokine levels of GM-CSF, IFN-α, IFN-γ, IL-2, IL-4, IL-5, IL-6, IL-9, IL-10, IL-17A, IL-12, and TNF-α is not neutrophils. The significant decrease in cytokine levels indicates an immunomodulatory effect of SARS-CoV-2 infection on cytokine production by PBMCs and neutrophils in COVID-19 ICU patients. The significant increase in IL-2 and IL-5 production by PBMCs may indicate the involvement of PBMCs in the increased IL-2 and IL-5 levels observed during the cytokine storm in COVID-19 ICU patients. We detected a decreased Th1-cytokine bias in serum, PBMCs, and neutrophils of COVID-19 ICU patients; Th1-Th2 cytokine ratios show a lower inflammatory cytokine bias in 17 of the potential combinations for PBMCs, and neutrophils in COVID-19 ICU patients. Our results suggest a downregulatory effect of SARS-CoV-2 infection on the production of cytokines by PBMCs and neutrophils. These findings may lead to new options for immune interventions to repair antiviral immune responses associated with enhanced viral clearance in SARS-CoV-2 infected patients. Limitations of this study include the small sample size and the fact that the subjects studied were not a homogeneous group. Studies on a larger homogenous group would support the decisions of this study.

## Figures and Tables

**Figure 1 microorganisms-10-02194-f001:**
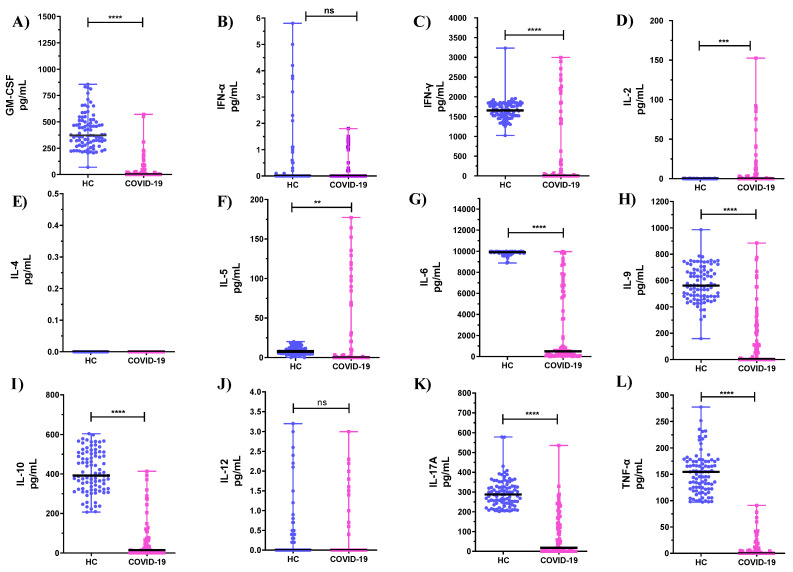
Cytokine profiles of PHA-stimulated PBMCs from healthy subjects and COVID-19 ICU patients (**A**–**L**). ** *p* < 0.01, *** *p* < 0.001, **** *p* < 0.0001 refers to the statistical differences. HC: healthy controls; ns: not significant.

**Figure 2 microorganisms-10-02194-f002:**
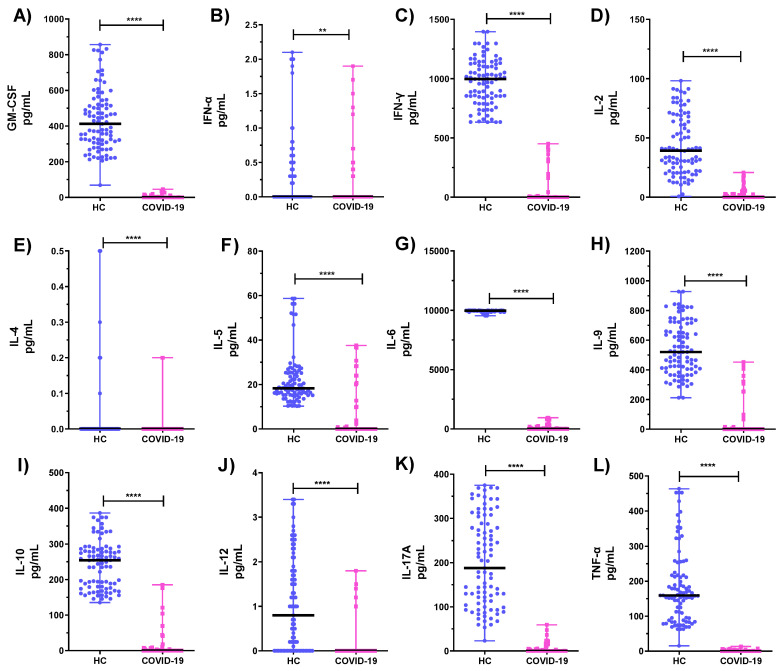
Cytokine profiles of neutrophils from healthy subjects and patients in the COVID-19 intensive care unit (**A**–**L**). ** *p* < 0.01, **** *p* < 0.0001 refers to the statistical differences. HC: healthy controls; ns: not significant.

**Figure 3 microorganisms-10-02194-f003:**
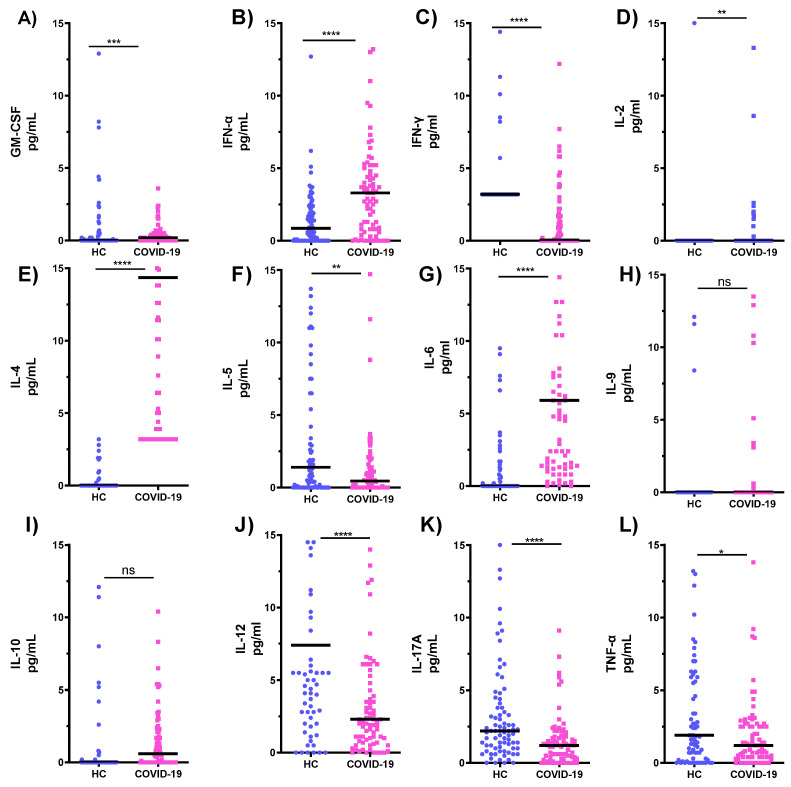
Cytokine profiles in serum samples from healthy subjects and patients in the COVID-19 intensive care unit (**A**–**L**). * *p* < 0.05, ** *p* < 0.01, *** *p* < 0.001, **** *p* < 0.0001 refers to the statistical differences. HC: healthy controls; ns: not significant.

**Table 1 microorganisms-10-02194-t001:** Deceased and transferred COVID-19 ICU patients stratified by gender and age.

	Deceased	Transferred	OR (95% CI)	*p*-Value
Study Sample	10 (11.4%)	78 (88.6%)	0.97 (0.94–1.00)	0.029
Gender				0.493
F	4 (40.0%)	23 (29.5%)		
M	6 (60.0%)	55 (70.5%)	1.59 (0.41–6.18)	
Mean age, years	60.9	52.8	0.95 (0.91–1.00)	0.112

OR: Odd ratio; 95% CI: 95% confidence interval.

**Table 2 microorganisms-10-02194-t002:** Clinical data of deceased and transferred COVID-19 ICU patients.

	Deceased	Transferred	OR (95% CI)	*p*-Value
Fever:				0.009
No	5 (50.0%)	12 (15.4%)		
Yes	5 (50.0%)	66 (84.6%)	5.50 (1.38–21.95)	
Cough:				<0.0001
No	9 (90.0%)	20 (25.6%)		
Yes	1 (10.0%)	58 (74.4%)	26.1 (3.11–219.21)	
Pneumonia:				0.608
No	0 (0.00%)	2 (2.56%)		
Yes	10 (100%)	76 (97.4%)	1.46 (0.06–32.50)	
Other Infections:				0.0005
No	8 (80.0%)	20 (25.6%)		
Yes	2 (20.0%)	58 (74.4%)	11.6 (2.27–59.27)	

95% CI: 95% confidence interval.

**Table 3 microorganisms-10-02194-t003:** Laboratory data of deceased and transferred COVID-19 ICU patients.

	Deceased	Transferred	OR (95% CI)	*p*-Value
HB	11.6 (2.41)	15.9 (19.1)	1.12 (0.84–1.50)	0.063
WBC	15.5 (8.26)	12.1 (6.89)	0.94 (0.86–1.02)	0.231
Lymphocytes	1.06 (0.82)	0.92 (0.57)	0.73 (0.29–1.84)	0.618
PLT	205 (113)	259 (107)	1.00 (1.00–1.01)	0.180
Lactate	6.65 (9.25)	2.24 (1.95)	0.83 (0.72–0.97)	0.191
LDH	1145 (1476)	850 (1469)	1.00 (1.00–1.00)	0.584
D-Dimer	11521 (19281)	1601 (2427)	1.00 (1.00–1.00)	0.161
CRP	122 (53.2)	174 (96.5)	1.01 (0.99–1.02)	0.220
Na	139 (6.11)	137 (6.34)	0.95 (0.85–1.06)	0.374
K	4.37 (0.65)	4.17 (0.68)	0.64 (0.25–1.69)	0.369
Acid/Base:				0.316
Abnormal	7 (70.0%)	38 (48.7%)	─	
Normal	3 (30.0%)	40 (51.3%)	2.37 (0.59–12.3)	
Ferritin	6272 (13632)	1011 (772)	1.00 (1.00–1.00)	0.388
Urea	10.8 (4.11)	8.62 (5.86)	0.94 (0.85–1.04)	0.154
Creatinine	149 (90.7)	98.8 (70.8)	0.99 (0.99–1.00)	0.120
ALT	163 (379)	99.7 (344)	1.00 (1.00–1.00)	0.627
AST	318 (763)	197 (1057)	1.00 (1.00–1.00)	0.663
TBill	26.2 (38.7)	13.2 (14.6)	0.98 (0.96–1.00)	0.318
Trop	34.2 (32.5)	14.7 (50.6)	0.99 (0.98–1.00)	0.119
BNP	12909 (21201)	1859 (3818)	1.00 (1.00–1.00)	0.462

95% CI: 95% confidence interval; HB: hemoglobin; WBC: white blood count; PLT: platelets; LDH: lactate dehydrogenase; D-Dimer: disseminated intravascular coagulation; CRP: C-reactive protein; Na: sodium; K: potassium; ALT: alanine amino transferase: AST: aspartate aminotransferase; TBil: total bilirubin; Trop: troponin; BNP: B-type natriuretic peptide.

**Table 4 microorganisms-10-02194-t004:** Ratios of pro-inflammatory to anti-inflammatory cytokines in COVID-19 ICU patients and healthy subjects.

Th1/Th2	PBMCs		Neutrophils		Serum	
	COVID-19	HC	COVID-19	HC	COVID-19	HC
IL-2/IL-4	48.23	12.56	67.565	11.011	0.0067	0.0007
IL-2/IL-5	1.53	3.55	0.253	0.493	0.8571	0.0001
IL-2/IL-9	0.475	0.02	0.009	0.05	0.8888	0.0002
IL-2/IL-10	2.7602	0.04	0.081	0.33	0.48	0.0001
TNF/IL-4	33.1	342.6	22.5	92.6	0.1179	0.4724
TNF /IL-5	2.4	58.20	1.8	38.6	1.5	0.7894
TNF/IL-9	0.326	0.773	0.2	0.301	1.5	1.7142
TNF/IL-10	0.372	0.174	2.7	0.897	0.84	0.3208
IL-6/IL-4	792.6	7,834.8	327.1	5,144.6	4.3	0.0866
IL-6/IL-5	58.2	1143.4	122.6	451.20	54.7	0.1447
IL-6/IL-9	7.80	17.844	23.6	18.317	53.9	0.3143
IL-6/IL-10	13.92	45.452	239.3	40.865	30.64	0.0588
IL-12/IL-4	0.283	0.00005	296.5	328.10	0.365	3.968
IL-12 /IL-5	0.021	0.00001	0.084	0.0990	4.643	6.632
IL-12/IL-9	0.003	0.000005	0.0003	0.0021	4.814	14.4
IL-12/IL-10	0.016	0.000003	0.005	0.0067	2.6	2.695
IFN-α/IL-4	0.347	0.8073	0.075	0.166	0.258	0.126
IFN-α/IL-5	0.025	0.1096	0.0038	0.017	3.286	0.211
IFN-α/IL-9	0.0003	0.0009	0.00005	0.001	3.407	0.457
IFN-α/IL-10	0.0011	0.025	0.00056	0.001	1.84	0.085
IFN-γ/IL-4	274.2	1177.3	443,600	533.6	0.124	3.772
IFN- γ/IL-5	20.1	209.9	161.7	43.8	1.572	6.303
IFN- γ/IL-9	2.7	3.0	1.6	1.8	1.629	13.69
IFN- γ/IL-10	1.7	4.2	17.3	4.0	0.88	2.561

HC: healthy controls.

## Data Availability

Data supporting reported results can be obtained from the corresponding author upon reasonable request.

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
