# Peer review of "Influence of SARS-COV-2 Infection on Cytokine Production by Mitogen-Stimulated Peripheral Blood Mononuclear Cells and Neutrophils in COVID-19 Intensive Care Unit Patients"

_microorganisms, 2022, doi:10.3390/microorganisms10112194_

Round 1

Reviewer 1 Report

The manuscript “Influence of SARS-CoV-2 Infection on Cytokine Production by Mitogen-Stimulated Peripheral Blood Mononuclear Cells and Neutrophils in COVID-19 Intensive Care Unit Patients” evaluated the profile of several cytokines (i.e. GM-CSF, IFN-α, IFN-γ, IL-6, IL-9, IL-10, IL17A and TNF-α) in the circulating system of COVID-19 patients admitted to the intensive care unit.

The manuscript is good, the results are well described and the author’s aims have been achieved. However, below some little suggestions to improve the manuscript:

-        English revision is suggested.

-        Line 54-56, Introduction: Please provide a reference to this sentence.

-        Materials and Methods, Study population section: It could be better to indicate the type of samples has been collected from patients (i.e. blood and respiratory samples) when the population is presented (line 76-79). Similarly, the same information should be added for the group of healthy donors (line 80-81).

-        Materials and Methods, 2.2 section: please provide a reference to the sentence of line 118-120.

-        Materials and Methods, 2.3 section: what was the final volume of whole blood used to isolate PBMCs? Similarly, what was the final volume of whole blood used to isolate neutrophils?

-        Results: for all the p-value indicated in the results section indicate the type of statistical analysis used.

-        Please describe the methods used to collect serum samples.

-        Figure 1 and Figure 2: the significance bars should be placed outside the graphs. Pay attention, they are overlapped with plots.

Author Response

Responses to Reviewer 1 Comments

Comment 1: The manuscript is good, the results are well described and the author’s aims have been achieved. However, below some little suggestions to improve the manuscript

Response: We thank the reviewer for the very helpful comments and suggestions. We have done our best to address the concerns of the reviewer. 

Comment 2: English revision is suggested.

Response: The paper has been revised to improve the language.

Comment 3: Line 54-56, Introduction: Please provide a reference to this sentence.

Response: We have added a reference [6] for this sentence in the introduction section, 2nd page, 2nd paragraph, line 5.

Comment 4: Materials and Methods, Study population section: It could be better to indicate the type of samples has been collected from patients (i.e. blood and respiratory samples) when the population is presented (line 76-79). Similarly, the same information should be added for the group of healthy donors (line 80-81).

Response: We have added the type of samples collected (blood and respiratory samples) to the Materials and Methods, section 2.1., lines 8-11. The sentence now reads " Nasopharyngeal swab samples were collected for respiratory investigations and blood samples were collected for laboratory investigations and isolation of PBMCs and neutrophils from each subject (COVID-19 patients and healthy controls)."

Comment 4: Materials and Methods, 2.2 section: please provide a reference to the sentence of line 118-120.

Response: We have added a reference [22] to the Materials and Methods, section 2.2., line 10.

Comment 5: Materials and Methods, 2.3 section: what was the final volume of whole blood used to isolate PBMCs? Similarly, what was the final volume of whole blood used to isolate neutrophils?

Response: The final volume of whole blood used to isolate PBMCs and neutrophils was 4 ml each.

Comment 6: Results: for all the p-value indicated in the results section indicate the type of statistical analysis used.

Response: We used an unpaired t-test for all the p-values indicated in the Result section. In the Materials and Methods section 2.7., lines 6-7 the sentence now reads "An unpaired t-test was used to calculate the p values."

Comment 6: Please describe the methods used to collect serum samples.

Response: To obtain serum, blood was collected in a serum tube (containing a clot activator) The tubes were then gently inverted several times and left in an upright position for 30 minutes at room temperature to allow the blood to clot. We then centrifuged the tubes in a refrigerated centrifuge at 2,000 x g for 10 minutes and the supernatant serum was aspirated.

Comment 7: Figure 1 and Figure 2: the significance bars should be placed outside the graphs. Pay attention, they are overlapped with plots.

Response: We appreciate this important observation by the reviewer. The figures have been changed accordingly.

Reviewer 2 Report

Very interesting article.

1- The article investigates cytokine production in intensive care unit patients with SARS-COV 2. Specifically, it studies the increased production of numerous cytokines (granulocyte-macrophage colony-stimulating factor (GM-CSF), interferon (IFN)-α, IFN-γ, interleukin (IL)-2, -4, -5, -6, -9, -10, -12, 20, -17A, and tumor necrosis factor (TNF)-α)by neutrophils and mononuclear cells present in peripheral blood. The study is performed by exploiting specific anti-cytokine antibodies.

2- At present, it is certainly not new that COVID-19 produces a "cytokine storm"(often responsible for sepsis), so it would be incorrect to speak of true novelty. Certainly, it is a study that helps to confirm the knowledge currently found and especially corroborates current theories about immune defenses against COVID-19. However, we cannot say that it definitively addresses a specific gap in the field.

3- As I said, the  "cytokine storm" topic has already been widely addressed. However, what I think is interesting about this article is, in terms of the results, the correlation between the reduction in the production of certain cytokines in peripheral blood. This result makes us realize that SARS-COV2 can stimulate some cells rather than others. specifically neutrophils seem to be "inhibited "in their cytokine production. One would have to ask whether this reduction goes hand in hand with an increase in opportunistic bacterial infections. Another interesting group of controls would be those who developed nosocomial infections during hospitalization.  

4- Regarding the methods, I have nothing to add about the choice of instrumentation for cytokine and virus detection. I appreciated the choice of patients in that both comorbidities and other viral co-infections were considered. I don't think I would have chosen different controls. A healthy population is a shareable control. It would be interesting to reproduce a statistic by age (dividing subjects into groups by age).  

5- I appreciated the graphical representation of the results.

6- The conclusions are well set and, in my opinion, meet the purpose of the research.

7- I suggest that you expand the bibliography by adding these articles:

- The first article deals with excessive cytokine release from COVID 19 resulting in myocardial damage:

Myocardial Pathology in COVID-19-Associated Cardiac Injury: A Systematic Review. Diagnostics (Basel, Switzerland), 11(9), 1647. https://doi.org/10.3390/diagnostics11091647

- In contrast, the second article proposes a case of death after the administration of the anti-covid19 vaccine. This means that vaccine administration can also lead to an elevated cytokine storm:

Death after the Administration of COVID-19 Vaccines Approved by EMA: Has a Causal Relationship Been Demonstrated? Vaccines, 10(2), 308. https://doi.org/10.3390/vaccines10020308

Author Response

Responses to Reviewer 2 Comments

Comments 1 and 2:

  1. The article investigates cytokine production in intensive care unit patients with SARS-COV-2. Specifically, it studies the increased production of numerous cytokines (granulocyte-macrophage colony-stimulating factor (GM-CSF), interferon (IFN)-α, IFN-γ, interleukin (IL)-2, -4, -5, -6, -9, -10, -12, 20, -17A, and tumor necrosis factor (TNF)-α) by neutrophils and mononuclear cells present in peripheral blood. The study is performed by exploiting specific anti-cytokine antibodies.
  2. At present, it is certainly not new that COVID-19 produces a "cytokine storm"(often responsible for sepsis), so it would be incorrect to speak of true novelty. Certainly, it is a study that helps to confirm the knowledge currently found and especially corroborates current theories about immune defenses against COVID-19. However, we cannot say that it definitively addresses a specific gap in the field. 

Response: We respect and appreciate the reviewer's comments; we submit that our manuscript provides useful novel data on the downregulatory effect of SARS-CoV-2 infection on the production of cytokines by PBMCs and neutrophils.

Comment 3: As I said, the "cytokine storm" topic has already been widely addressed. However, what I think is interesting about this article is, in terms of the results, the correlation between the reduction in the production of certain cytokines in peripheral blood. This result makes us realize that SARS-COV2 can stimulate some cells rather than others. Specifically, neutrophils seem to be "inhibited "in their cytokine production. One would have to ask whether this reduction goes hand in hand with an increase in opportunistic bacterial infections. Another interesting group of controls would be those who developed nosocomial infections during hospitalization.  

Response: We thank the reviewer for highlighting the specific novel information on the “inhibition” of cytokine production by neutrophils and the implication that cells other than PBMCs and neutrophils contribute to the cytokines measured in the blood. We agree that it would be very interesting to note whether the observed effects on cytokine production are associated with nosocomial bacterial infections; we propose to address this in our next project.

Comment 4: Regarding the methods, I have nothing to add about the choice of instrumentation for cytokine and virus detection. I appreciated the choice of patients in that both comorbidities and other viral co-infections were considered. I don't think I would have chosen different controls. A healthy population is a shareable control. It would be interesting to reproduce a statistic by age (dividing subjects into groups by age).  

Response: The reviewer makes a very valid and important point; it would certainly be of interest to study age-related cytokine responses. However, in the current study most of our subjects were elderly patients.

Comment 5: I appreciated the graphical representation of the results.

Response: We thank the reviewer for this comment.

Comment 6: The conclusions are well set and, in my opinion, meet the purpose of the research.

Response: We thank the reviewer for this comment.

Comment 7: I suggest that you expand the bibliography by adding these articles:

Myocardial Pathology in COVID-19-Associated Cardiac Injury: A Systematic Review. Diagnostics (Basel, Switzerland), 11(9), 1647. https://doi.org/10.3390/diagnostics11091647

Death after the Administration of COVID-19 Vaccines Approved by EMA: Has a Causal Relationship Been Demonstrated? Vaccines, 10(2), 308. https://doi.org/10.3390/vaccines10020308

Response: We thank the reviewer for these suggestions. We have added these references to the list (references 3 and 5).
